# Fears, Reassurance, and Milestones: A Twitter Analysis around World Prematurity Day during the COVID-19 Pandemic

**DOI:** 10.3390/ijerph182010807

**Published:** 2021-10-14

**Authors:** Kathy McKay, Elizabeth O’Nions, Sarah Wayland, David Ferguson, Eilis Kennedy

**Affiliations:** 1Public Health, Policy and Systems, Institute of Population Health, University of Liverpool, Liverpool L69 3GL, UK; 2Research and Development Unit, Tavistock and Portman NHS Foundation Trust, London NW3 5BA, UK; ekennedy@Tavi-Port.nhs.uk; 3Research Department of Clinical, Educational & Health Psychology, University of College London, London WC1E 6BT, UK; e.onions@ucl.ac.uk; 4School of Health, University of New England, Armidale 2350, Australia; swaylan2@une.edu.au; 5Formerly, Faculty of Transdisciplinary Innovation, University of Technology Sydney, Sydney 2007, Australia; dave.x.ferguson@gmail.com

**Keywords:** premature birth, preterm birth, NICU, maternal mental health, paternal mental health, child development, COVID-19, Twitter

## Abstract

Preterm birth (birth <37 completed weeks’ gestation) is common, affecting 10.6% of live births globally (nearly 15 million babies per year). Having a new baby admitted to a neonatal unit often triggers stress and anxiety for parents. This paper seeks to explore experiences of preterm birth via Twitter. The intermingling of COVID-19 restrictions and World Prematurity Day allows for an understanding of both the additional stresses incurred as a consequence of the pandemic and the more “everyday” experiences in the NICU and beyond. The content analysis of the data included 3161 tweets. Three themes were identified: 1. COVID-19 was not the only trauma; 2. Raising awareness, especially World Prematurity Day; and, 3. Baby milestones. These themes highlight the multi-level challenges faced by parents of premature babies and the healthcare professionals involved in their care. The COVID-19 pandemic and the consequent restrictions imposed on parents’ contact with their babies have resulted in immense emotional strain for families. The reported COVID-19 pandemic “baby blind spot” appears to particularly impact this group of babies. Improved understanding of the lived experiences of preterm babies and their families should inform greater awareness and improved support.

## 1. Introduction

Preterm birth (birth <37 completed weeks’ gestation) is common, affecting 10.6% of live births globally (nearly 15 million babies per year) [1]. The duration of a preterm infant’s stay on a neonatal unit varies considerably depending on their health, birthweight, and gestational age at birth. In the U.K., the average length of stay on a unit for preterm babies is seven days, but for the minority of babies born at <28 weeks, it is 92 days, and for those born at 28–31 weeks, 44 days [2]. These infants have a very different start in life compared to babies who can be discharged immediately to the care of their parents.

Having a new baby admitted to a neonatal unit often triggers extreme stress and anxiety for parents. Even for babies who are medically stable, preterm birth and/or hospitalisation is a traumatic event, triggering guilt, distress, and depression in parents [3,4,5,6]. Parents of preterm babies are reported to show higher levels of PTSD symptoms compared to term-born babies [7]. The severity of their PTSD has been linked to the child’s state at birth: parents whose babies are most unwell suffer the most acute psychological impact [8].

A particular challenge for parents of hospitalised neonates is not being able to be physically present with their baby all the time [4,5]. Because only a minority of units offer parents the option of “rooming in”, hospitalisation usually necessitates extended periods of separation, triggering acute distress and, in some cases, feelings of disconnection from the infant [9,10]. This may be particularly distressing for parents of the most unwell babies, whose condition can deteriorate rapidly and unexpectedly.

Physical proximity to the baby is important because early interactions with and physical closeness to neonates are important in stimulating the formation of parent-infant bonds [11]. A growing body of evidence suggests that augmenting parent-infant contact for hospitalised neonates may be key to fostering better long-term outcomes for parents and infants and may buffer the adverse impacts of premature birth on development [12].

The timing of parental contact may also be important. For preterm infants, skin-to-skin contact plays an important role in facilitating the maturation of the parasympathetic arm of the infant’s autonomic nervous system, which usually occurs at around 33 weeks gestation [12]. One hour per day of “Kangaroo” care, where the infant is placed unclothed on the parent’s chest, leads to improved stress regulation and developmental outcomes for preterm babies in the short and longer term [12,13]. Positive outcomes are also found for maternal mental health and parent-infant interaction [14].

Recent advances in neonatal care practices (e.g., family-centred care) [15] have highlighted the stress-buffering effects of involving parents in their infant’s care by allowing them to assist with basic care tasks in the NICU [16]. According to the U.K. charity Bliss, this helps “parents to feel like parents—which may be key for their own perceptions of attachment to their baby” ([17] (p. 6)). Engaging parents in this way may also reduce the likelihood that severe distress about their baby’s fragility triggers emotional and physical withdrawal from the baby [18,19].

The ongoing COVID-19 pandemic has led to extreme pressure on health services, leading to widespread changes in hospital policy to reduce infection risk to patients and staff. According to Hynan [20], “The prevailing values and goals of any NICU have been replaced by the urgent need to curtail this pandemic” (p. 985). Concerns about the immediate physical health of babies and staff have taken priority over concerns about mental health or longer-term outcomes [20]. This has led to restrictions on parental access to neonatal units. Restrictions vary, but in some cases have included limits of 2 h per day on parental visits, which may apply even when the baby is acutely ill, and a requirement that only one parent can be on the unit at a time, or only one nominated parent can visit [21]. Parents of multiple births are also, in some cases, asked to divide their allocated time between them [21].

Although some parents report satisfaction with limitations on visits to protect the health of their baby, others find restrictions deeply upsetting [22]. Parents report that access restrictions have adversely impacted the number of visits, time available for parent-infant bonding, participation in care, and breastfeeding. Anecdotally, one parent reported having to choose between whether to spend time cuddling their infant or learning about their care [22].

Policies that require only one parent on the unit at a time mean that parents have struggled to cope with the distress of seeing their hospitalised baby without the support of their partner [22]. Others have expressed concerns over whether their babies can recognise them due to the need to wear masks at the cot-side [22]. Subjectively, this has led some parents to feel as though they are not able to effectively support or care for their child [22]

Work investigating the impact of COVID-19 on neonatal services has highlighted considerable local variation in policy [22]. According to Dr. Helen Mactier, president of the British Association of Perinatal Medicine (BAPM), “In most cases, a rule made up for other [hospital departments] gets blanket applied” [23]. Bliss, the U.K. charity for babies born premature or sick, has argued that these restrictions are potentially devastating for those with babies who have only have days or hours left to live. They are deeply worrying for all parents because, according to NHS England, “high quality neonatal care must include a substantive role for parents in the care of their baby […] Parents are not bystanders… but perform an active role as members of the clinical team” ([24] (p. 17)).

For preterm infants, reductions in opportunities to offer kangaroo care may lead to the sensitive window for maturation of the autonomic nervous system being missed. Parent mental health may also suffer: in a unit that had strict limitations on parent-infant contact pre-COVID-19, as many as 60% of mothers and 47% of fathers reported significant PTSD symptoms [25]. These findings lend support to the perspective echoed by the BAPM: “The mother and her newborn are a biological entity and should have unrestricted contact when admission to a NNU is unavoidable” ([26] (p. 14)). Restrictions to parent-infant contact as a consequence of the COVID-19 pandemic are therefore likely to have adverse repercussions for families in the short and longer term; however, at present, little is known about the impact of these policies on parents.

This paper seeks to explore parents’ and healthcare professionals’ experiences of caring for premature babies in the NICU during the COVID pandemic via Twitter. The intermingling of COVID-19 restrictions and World Prematurity Day during the time in which the tweets were collected allows for a better understanding of how dealing with the pandemic exacerbated the stresses attached to premature birth, as well as the more “everyday” experiences in the NICU.

## 2. Materials and Methods

D.F. used Twitter API software, as well as the R Studio software and R package retweet, to collate and download tweets daily before adding them to an R data file. Tweets sent between 24 October 2020 and 30 November 2020 were extracted in order to analyse the narratives around premature babies in a NICU in real time. This process of collecting tweets is in line with previous research [27,28]. The research team chose hashtags both relevant to the experience of premature birth and the cohort of parents and healthcare professionals in a NICU during the COVID-19 pandemic, as well as those that would allow for a wide range of tweets to be identified. The hashtags searched were #preterm, #parentalaccess, #neonatalcare, #nicu, #maternalhealth, and #neonatalhealth. These hashtags represent the search terms joined with “OR”. Tweets were extracted daily and added to an R data file that can be backed up and re-read for subsequent analysis.

Altogether, 3161 tweets were collected and included in the analysis. As well as the text of the tweet, the time and date it was sent, tweet ID, handle, link, number of favourites, number of retweets, mentions (of other Twitter users), first search term, number of search hits, potential retweet, hashtags, and tweet month were also collected.

K.M. undertook content analysis. She identified concepts, categories, and themes in line with Bengtsson’s process [29]. Tweets were then coded around positive and negative sentiments [30]. Positive sentiments tended to be grounded in experiences that felt supportive or helpful. Negative sentiments were grounded in distressing or frustrating experiences. K.M. then checked the themes with the other authors.

No ethical approval was sought for this data collection as, in line with Pedersen and Lupton [31] and Fiesler and Proferes [32], only “spontaneously generated” tweets in the public domain were collected. This meant that tweets from closed or private accounts could not be included, and no tweets were elicited by the authors. To help ensure anonymity, handles are not included with any tweets, but dates have been left on the tweets used in the paper. In addition, photos were often included in tweets from parents or if staff in a NICU were celebrating a milestone (for example, a baby gaining weight or being well enough to leave the NICU). However, as an extra layer of anonymisation, links to any photos have not been included even though all tweets were public.

## 3. Results

The analysis of the data included 3161 tweets. The tweets were bound to the month that World Prematurity Day falls in, with the majority of the tweets sent in the week surrounding the 17 November 2020. Tweets were sent by parents, healthcare professionals, and via accounts of maternity hospitals and clinics.

Three themes were identified (see Table 1 for exemplar quotes):COVID-19 was not the only trauma;Raising awareness, especially World Prematurity Day;Baby milestones.

**Table 1 ijerph-18-10807-t001:** Themes and exemplar quotes.

Theme	Exemplar Quotes
	Parent	Healthcare Professional
COVID-19 was not the only trauma	Women, children and their families are being let down time and time again in this pan-demic.This will affect us all for years to come.It needs to stop happening.#maternalhealth #childrenshealth #familyhealth #motherhood #supportfamilies (29 October 2020)Waiting for my son to get a #COVID test and waiting for the results is a hell I wouldn’t wish on anyone. I haven’t been in this bad of a place mentally since he was in the #NICU (28 November 2020)	Did you know?More than 15 million babies around the world are born prematurely every year, and prematurity is the leading cause of death in children under five years of age.#nicu #nicubaby #preemie #weeker #niculife #nicumom #preemiestrong #preemiebaby #baby (3 November 2020)Keep being told about newborns coughing/sneezing once or twice in #NICU on report these days.Me thinks that it always was this way but #COVID19 has us all a little hypervigilant!To swab or not to swab. That is the question! (22 November 2020)
Raising awareness, especially World Prematurity Day	7 years ago I was counting down 6 weeks to my Xmas due date. Little did I know that in less than 10 days we’d be in the #NICU—this amazing little guy is incredible… but still prefers to sleep in the middle of people noise tho 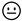 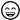 #WorldPrematurityDay2020 (17 November 2020)	Did lockdowns lower premature births? Dutch researchers say the “impact was real,” adding to hopes that doctors will learn more about factors contributing to preterm birth.#prematurebabies #preterm #birth #lockdown (26 November 2020)
Baby milestones	The last 11 days have been an emotional rollercoaster. I’ve came out of the hospital this evening after seeing [baby] and I see this beautiful double rainbow. It reminded me of the quote that after a storm comes a rainbow. My rainbow is coming. 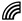 #nicubaby #NICU #preemiestrong (27 October 2020)	Today I helped a very anxious mummy hand express for the first time, she didn’t think she’d be able to do it but she did and managed to express 0.3 mL and it was brilliant to be a part of! 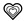 #NICU #HandExpressingMilk #BreastFeeding (4 November 2020)

### 3.1. COVID-19 Was Not the Only Trauma

Tweets about the difficulties and fears COVID-19 had and continued to present when taking care of premature babies and their mothers came from across the globe, as countries struggled with the pandemic. COVID-19 not only affected the care able to be offered but increased maternal mortality risk:

*#Maternal care hard hit by #Covid-19 pandemic in #Bangladesh #COVID**-19 #maternalhealth #maternalmortality #COVID19 #coronaravirus* (13 November 2020).

There was a sense that healthcare providers wanted to reassure parents that they would be able to care for their babies despite the risks attached to COVID-19:

*#COVID19 wont stop our global #NICU ASP study in S Africa! toolkits arrived in Joburg!! We will*
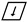
*mortality together @inspired2leadQH @PedsPharmD @dinimawela @NSchellack @AngPharmID @OSUGlobal @osu_pharmacy* (30 October 2020).
*Despite the challenges presented by #COVID-19, CMMB is continuing its work to support mothers and newborns in #Haiti through antenatal care, skilled deliveries, and other services.*

*Learn about our maternal health projects:*
*#WomensHealth #MaternalHealth* (6 November 2020)
*We have also relayed vital updates to 800,000 frontline health workers across India in collaboration with The Ministry of Health Family Welfare (@MoHFW_INDIA)*
*#COVID #COVID19 #Corona #healthcare #health #Virtualclinic #pandemic #maternalhealth #childhealth #mHealth #HealthTech* (23 November 2020).

The risks attached to both mother and baby from COVID-19 were outlined as more knowledge was gained throughout the pandemic:

*“Although the absolute risks for severe COVID-19–associated outcomes among women were low, pregnant women were at significantly higher risk for severe outcomes compared with nonpregnant women” from @CDCgov #COVID19 #HealthEquity #MaternalHealth* (3 November 2020)*Pregnant individuals have a higher risk of severe illness from #COVID19 than non-pregnant individuals. CDCgov lays out the steps of how to keep yourself and your newborn safe from contracting COVID-19: #MaternalHealth #InfantHealth* (5 November 2020)

Some tweets mentioned specific cases where a pregnant woman had died after contracting COVID-19 as a way to demonstrate that the threat was real and caution was needed:

*‘I can’t fail Mary’: the bereaved man fighting for pregnant women threatened by Covid #schools #CloseTheSchools #edutwitter #masksforschools #maternalhealth #PutSchoolsInLockdown #teachers #Friday13th* (13 November 2020)*B.C. woman on life support after catching COVID-19 while pregnant* via *@CBCTheNational*

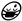

*#COVID19 #COVID__19 #Covid #pandemic #epidemiology #pregnancy #healthcare #Health #maternalhealth #publicmedia #ViewFromCanada* (17 November 2020)

For these reasons, many tweets about premature births and babies and COVID-19 focused on the hospital protocols put in place to protect both patients and staff. These tweets either focused on practical safety measures:

*Although hospital staff always follow very strict cleaning precautions in the #NICU, understanding safety measures you can take might help ease your mind. Here are some #tips for keeping yourself and your #NICU baby safe from #respiratory viruses. [link]* (28 October 2020)

Alternatively, these tweets included photos of babies as a way to illustrate the human side to the safety measures:

*ADORABLE: With extra safety measures put in place due to the coronavirus, the #NICU team in #Detroit did not disappoint with the most adorable pumpkin costumes [link] for their tiny babies. [link to photo]* (29 October 2020)

One of the most lingering aspects of the impact of COVID-19 in terms of the safety measures required was the limitation of time a parent of a preterm baby could spend with their child in the NICU, and often limiting this time to the biological mother. While these safety measures were understood within the pandemic context, this did not necessarily mitigate the distress such limitations caused parents who were already dealing with the stresses attached to a “normal” premature birth:

*“And then, the coronavirus hit.”…. Suffering from postpartum and perinatal anxiety and depression was hard enough pre-COVID-19. The pandemic brings along with it many questions that are new territory. [link] #maternalhealth [name] [link]* (28 October 2020)*Dr. Deena Hinshaw admits the restrictions will be very difficult “but these temporary measures are being implemented to help reduce exposure and spread of the virus in AHS facilities.” #COVID19AB #Hinshaw #Visitation #Hospitals #NICU* (26 November 2020)

Expressing concern that limiting a parent’s time with their premature baby would be harmful to both, media reports linked data around the importance of contact and attachment for babies in this early developmental phase, as well as for the long-term mental wellbeing of the mother:

*So proud of @TrudiSene1 highlighting the stark truths of how the lockdown has and will continue to affect women in the perinatal period @BBCNews #PMH #maternalhealth* (31 October 2020)*More data that #COVID is having a disproportionate impact on #women. Increased levels of #HTN due to #lockdowns increasing risk for poor #maternalhealth outcomes.* (3 November 2020)

Healthcare workers, particularly nurses who worked in NICUs, spoke about how concerns about keeping the babies safe from COVID-19 had made everyone in the NICU more aware of any possible symptom:


*Keep being told about newborns coughing/sneezing once or twice in #NICU on report these days.*

*Me thinks that it always was this way but #COVID19 has us all a little hypervigilant!*
*To swab or not to swab. That is the question!* (22 November 2020)

However, this was also balanced with the reality that premature babies had an increased chance of dying compared to babies born at term. Exposure to COVID-19 was not the only risk they faced:


*Did you know?*

*More than 15 million babies around the world are born prematurely every year, and prematurity is the leading cause of death in children under five years of age.*
*#nicu #nicubaby #preemie #weeker #niculife #nicumom #preemiestrong #preemiebaby #baby* (3 November 2020)

Some tweets also managed to be more light-hearted amidst the extra pressures during the pandemic:

*Although it’s frustrating that #masks keep fogging up my equipment, it is nice delivering babies without the fear of getting amniotic fluid in my mouth #obgyn #ob #maternalhealth #PPE [link]* (23 October 2020)

In this section of the paper, we present those tweets relating to the lived experience of parents, who also tweeted extensively about their experiences of the NICU during the pandemic. These tweets were emotionally charged as they struggled with the time limits set on parents visiting their babies:

*Parents of sick and small newborns must not be treated as visitors, they are #caregivers and must have unrestricted access to #NICU* (14 November 2020)

They also worried about the longer-term impacts on mothers and children as the pandemic continued:


*Women, children and their families are being let down time and time again in this pandemic.*

*This will affect us all for years to come.*

*It needs to stop happening.*
*#maternalhealth #childrenshealth #familyhealth #motherhood #supportfamilies* (29 October 2020)

Some U.K. parents spoke about their experiences during lockdown. During the time the tweets were collected, the U.K. had already had one lockdown, and a second one was on the horizon. Feelings about the lockdown tended to differ depending on how well supported the parent felt and how well their baby was compared to the first lockdown. One mother saw how healthy her premature baby had become since the first lockdown:


*Lockdown 1 Vs. Lockdown 2*

*[name] a few days old in NICU wondering if he’d be ok and if we’d get him home Vs. now, loving life in a swing without a care in the world*
*#prematurityawarenessmonth #preemie #NICU #lockdown2uk #UKlockdown* (13 November 2020)

Another mother, however, viewed a second lockdown as further isolation from the support she needed and felt she was not alone in her concern:

*I am dreading a #SecondLockdown and I’m not the only one. New mums are already quite isolated due to the pandemic and baby classes are a Covid-safe lifeline. If they don’t run then we don’t see anyone. This is huge. @MMHAlliance #maternalhealth #UKlockdown* (31 October 2020)

Yet, still other parents whose children had been exposed to COVID-19 worried about test results, highlighting the layers of stress parents of premature babies were experiencing:

*Waiting for my son to get a #COVID test and waiting for the results is a hell I wouldn’t wish on anyone. I haven’t been in this bad of a place mentally since he was in the #NICU* (28 November 2020)

### 3.2. Raising Awareness, Especially World Prematurity Day

Parents, as well as hospitals and individual healthcare professionals, used the time around World Prematurity Day to raise awareness about premature birth and the long-term impacts on parents and children. Interestingly, research suggesting rates of premature births had appeared to decrease during the pandemic was shared. These tweets gained traction as they suggested that better understanding this phenomenon might help to provide better prevention and care:

*More evidence linking #COVID19 mitigation measures with reduction in #preterm births from the Netherlands @TheLancetPH @EBNEO* (27 October 2020)
*Did lockdowns lower premature births? Dutch researchers say the “impact was real,” adding to hopes that doctors will learn more about factors contributing to preterm birth.*
*#prematurebabies #preterm #birth #lockdown* (26 November 2020)

Alongside sharing information about the impacts of COVID-19, parents and healthcare professionals also shared their experiences of being in the NICU as a way to illustrate the different types of support that might be needed. Better support was often framed in terms of inclusion, where parents need to be considered as partners to the healthcare professionals in the NICU and where fathers should not be forgotten. These tweets seemed to especially gain traction as people’s feelings about the limitations on NICU visits were explored:

*Parents are increasingly regarded to be team members, not only in the team providing care for their infant(s) but also as team members of the broader team such as an entire NICU. #nicu #neonatal #parentengagement #preterm #prematurity [link]* (30 October 2020)*Both my babies had to spend time in the NICU when they were born. It was heartbreaking, exhausting, demanding—and I was in no physical state to manage it alone, as I was really quite ill myself both times. Dads have to be included not excluded #nicu* (12 November 2020)

In addition, tweets also sought to raise awareness of the precarious health of premature babies, where modern medicine was not always able to save these children. The grief attached to losing one or more babies needed to be explored:

*The Complex Emotions of Losing a Twin in the NICU #pegnancyandinfantlossawarenessmonth #infantloss #NICU* (26 October 2020)

In the week of World Prematurity Day, tweets around COVID-19 awareness began to appear again, couching the yearly reminder in pandemic terms:

*Particularly in times of #COVID19 we need to remember: parents and babies belong together #ZeroSeparation #WorldPrematurityDay2020. Join #WPD20Chat on 16 Nov 9AM EST/2PM GMT and contribute to the discussion and to raising awarness on #preterm birth @CochraneNeonate* (15 November 2020)

However, on the day itself, most of the tweets were based on lived experience. Parents remembered their experiences in the NICU. These tweets focused on the positives of survival:


*Today is #WorldPrematurityDay and a chance to reflect on my preterm birth journey which has shaped and strengthened our family. Great work being done by @WIRFWA*
*#worldprematurityday #worldpremday #pretermbirth #preterm #thewholeninemonths #WIRF #WIRFWA* (17 November 2020)*7 years ago I was counting down 6 weeks to my Xmas due date. Little did I know that in less than 10 days we’d be in the #NICU—this amazing little guy is incredible… but still prefers to sleep in the middle of people noise tho*
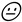

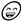
*#WorldPrematurityDay2020* (17 November 2020).

There were also tweets from adults who had been born prematurely. These tweets highlighted that their birth experience helped to shape their current career paths and philanthropic interests. These tweets provided hope that a baby who needed significant care at birth could grow into a happy adult:

*Having born premature and having my life being saved by a neonatal intensive care unit I really value their use. @UNICEFEthiopia works with @FMoHealth to combat neonatal deaths with all measures. #WorldPrematurityDay #HealthForAll #maternalhealth #NeonatalHealth* (17 November 2020)*When Shannon Sullivan learned she had spent the first 3 months of her life in Cedars-Sinai’s #NICU, she decided to dedicate her career to helping infants. “#Neonatology saved my life, and I need to pay it forward,” she says. #WorldPrematurityDay*
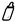
 (17 November 2020)

### 3.3. Baby Milestones

While COVID-19 and the precarious health of babies born preterm remained an ongoing preoccupation among parents and healthcare providers, positive milestones continued to be celebrated. These tweets highlighted that even with so much of the outside world changed for the adults around them, the milestones these babies achieved remained precious for everyone in the NICU. The emotion of these tweets resonated with the distress parents had expressed when their visiting times had been limited due to COVID-19 restrictions:

*The last 11 days have been an emotional rollercoaster. I’ve came out of the hospital this evening after seeing [baby] and I see this beautiful double rainbow. It reminded me of the quote that after a storm comes a rainbow. My rainbow is coming.*
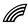
*#nicubaby #NICU #preemiestrong* (27 October 2020)*NICU Milestones: The First Diaper Change was the Hug I’d Been Waiting For #NICU* (30 October 2020)*No more respiratory support! #nicubaby #nicu #babyboy* (3 November 2020)*We are officially double birth weight! #nicu #nicubaby #babyboy* (7 November 2020).

Healthcare professionals in the NICU also celebrated when the babies in their care celebrated positive milestones, both during and after their care:

*Today I helped a very anxious mummy hand express for the first time, she didn’t think she’d be able to do it but she did and managed to express 0.3 mL and it was brilliant to be a part of!*
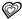
*#NICU #HandExpressingMilk #BreastFeeding* (4 November 2020)*I love when one of my #preemie #patients does so well at their check up that doctors use the terms #biggirl #heavy and #smart #childlife #infantmassage #support #coping #NICU #premature #allcaughtup* (9 November 2020)

## 4. Discussion

The aim of this paper was to explore the experiences of parents and healthcare professionals in the context of World Prematurity Day and the COVID-19 restrictions. Tweets are a useful way to gain real-time insights into stressful and emotionally charged experiences as they do not take long to craft and can be written spontaneously at the tweeter’s convenience. Further, the global nature of Twitter means that a person’s story can be shared across time zones and countries, which can lend strength to voices that may otherwise not be heard [27,33,34].

The tweets analysed in this paper highlight the layers of stress experienced by both parents of premature babies in a NICU and the healthcare professionals who care for them. It is clear that the COVID-19 pandemic has exacerbated anxieties for both staff and parents in neonatal units. In line with Muniraman et al. [22], the distress of parents having restrictions imposed on contact with their newborn babies was palpable in their tweets. Fears were expressed by both parents and healthcare staff about the consequences of this reduction in contact for the longer-term health and development of the babies. Inevitably this worry and anxiety regarding their babies’ wellbeing added further to the challenges parents were already facing with a knock on adverse impact on their own emotional wellbeing. These very real concerns needed, however, to be balanced with the risks to health as a consequence of COVID-19 spreading within the hospital. Tweets highlighting how COVID-19 severely affected pregnant women, with some women tragically dying, underlined the seriousness of the situation. A proportion of tweets, therefore, sought to raise awareness of the risk to pregnant women and also to offer reassurance to parents regarding hygiene procedures implemented within neonatal units to protect babies from the virus.

While tweets related to the COVID-19 pandemic and, in particular, the associated restrictions to parental visiting made clear the resulting distress for parents, they also emphasise the immense stress of premature birth itself, pandemic or not. Parents of premature babies are already at a higher risk of depression and feelings of guilt after the birth [3,4,5,6], as well as PTSD symptoms [7]. Being in the NICU is typically a physically stressful time for parents [17,21], adding further to the emotional stresses of having a baby whose survival is threatened.

While parental tweets primarily referenced emotional vulnerability experienced in the pandemic, many of the tweets from professional accounts (whether individual or institutional) referenced the practicalities of ensuring the physical safety of babies and protecting them from the virus. This is perhaps reflective of an ongoing tension within neonatal medicine where the health and safety of the babies can be prioritised at the expense of other considerations, such as the socio-emotional development of the baby.

Alongside all this, tweets also demonstrated how life went on in the NICU and beyond for families, even as the world changed significantly around them. Birthdays and milestones (such as meeting a goal weight or being discharged) continued to be celebrated. The way in which these tweets were generally crafted (photos, emojis, and many exclamation marks) demonstrated the importance of marking and celebrating such achievements for staff and families. These small, intimate celebrations were perhaps experienced as all the more important and meaningful in the context of the wider COVID-19 pandemic turbulence and its impact.

### Strengths and Limitations

There are several strengths and limitations of this paper. Twitter provides data at an individual level in real time captured across the world. Tweets can highlight similarities and differences within and across countries within a set time period. There is often little context to a tweet, and it can be uncertain how reflective one tweet might be of how a person truly feels, but they can demonstrate how emotions are being expressed in an unfiltered way. This topic also brought out people doing the same tweets several times at once about the same topic, which were little engaged with, and had to be filtered so as to not drown out other voices.

## 5. Conclusions

The findings from this paper illustrate the multi-level challenges faced by parents, healthcare workers, and the babies they care for during “normal” times, let alone during a global pandemic where the tensions between the requirement for infection control to protect the safety and survival of infants and the wish to promote infant wellbeing through parental contact are amplified. A COVID-19 “baby blind spot” has been identified where the needs of these most vulnerable members of society are often neglected [35]. Such a blind spot is perhaps all the more concerning for babies in neonatal care and their families who have already experienced so many challenges. Despite this, the resilience of families and their babies shines through in the acknowledgement and celebration of small, intimate, meaningful moments and milestones. A greater understanding of the lived experiences of preterm birth and the additional impact of the COVID-19 pandemic should inform improved support for families, babies, and the healthcare workers involved in their care.

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
