# Peer review of "Fears, Reassurance, and Milestones: A Twitter Analysis around World Prematurity Day during the COVID-19 Pandemic"

_ijerph, 2021, doi:10.3390/ijerph182010807_

Round 1
Reviewer 1 Report
Interesting approach to the subject. Gynecologists all around the world who counsel women in and prior to pregnancy are worried about the psychological impacts of the COVID pandemic on parents and children, especially after preterm birth. In the time of COVID-19-pandemic gathering information via internet is quite common and a pitfall for individual and thorough medical counseling. Alternative ways of spreading information, e.g. via twitter, is getting more and more important. This article summarizes the information about COVID-19 and psychological risks of prematurity which are spread via twitter at the world prematurity day. The study was performed in UK but international tweets were analyzed. Quotations as example are given to the themes. For the reader a structured way of presentation would facilitate the reading of the posts. I propose separating quotations of health care professionals and parents more clearly. Very good English, easy to understand, precisely worded Abstract – adequate Introduction – Good introduction to topic of prematurity and change of care due to the COVID-19-pandemic. Materials and method – Structured analysis of tweets and communication via twitter. 3161 statements were structured and analyzed. Results – three important themes were identified. This section is a first attempt to structure analysis of tweets. Still lack of structure, difficult to read Discussion – quite good, strengths and limitations are givenAuthor Response
Interesting approach to the subject. Gynecologists all around the world who counsel women in and prior to pregnancy are worried about the psychological impacts of the COVID pandemic on parents and children, especially after preterm birth. In the time of COVID-19-pandemic gathering information via internet is quite common and a pitfall for individual and thorough medical counseling. Alternative ways of spreading information, e.g. via twitter, is getting more and more important.
Thank you so much for this positive feedback and thoughtful reading of the paper.
This article summarizes the information about COVID-19 and psychological risks of prematurity which are spread via twitter at the world prematurity day. The study was performed in UK but international tweets were analyzed. Quotations as example are given to the themes. For the reader a structured way of presentation would facilitate the reading of the posts. I propose separating quotations of health care professionals and parents more clearly.
Thank you for this observation. We have made sure to signpost whether the tweet was written by a parent or healthcare professional throughout the manuscript and particularly highlighted this on page 7: “In this section of the paper, we present those tweets relating to the lived experience of parents, who also tweeted extensively about their experiences of the NICU during the pandemic”.
Very good English, easy to understand, precisely worded Abstract – adequate Introduction – Good introduction to topic of prematurity and change of care due to the COVID-19-pandemic. Materials and method – Structured analysis of tweets and communication via twitter. 3161 statements were structured and analyzed.
Thank you for your careful reading.
Results – three important themes were identified. This section is a first attempt to structure analysis of tweets. Still lack of structure, difficult to read
We hope that our changes above have made the results easier to follow. We have also added a table on pages 4-5 that clearly shows the themes and exemplar quotes (tweets).
Discussion – quite good, strengths and limitations are given
Thank you for this positive feedback.
Reviewer 2 Report
- What is your justification in terms of hashtags search form the twitter? You have not describe well, about location where the tweet has been recorded, why don't you put #england #british #BMP. I saw some tweet use hashtags #Bangladesh, also #ViewFromCanada
- How do you define caring a prematures babies in the NICU by the twitter. Please explain.
- Please add section about premature, NICU health services in new normal (during Covid19). Add more than 20 papers.
Author Response
What is your justification in terms of hashtags search form the twitter?
We apologise for any lack of clarification. Please see page 3 where additional justification of search terms is included: “The research team chose hashtags both relevant to the experience of premature birth and the cohort of parents and healthcare professionals in a NICU during the COVID-19 pandemic, as well as those which would allow for a wide range of tweets to be identified”.
You have not describe well, about location where the tweet has been recorded, why don't you put #england #british #BMP. I saw some tweet use hashtags #Bangladesh, also #ViewFromCanada
We apologise for any misunderstanding. The software identified the relevant tweets, not by location, but by use of the identified hashtags. Any hashtags that identified a specific location were written by the person sending the tweet. We cannot change the data and add in location hashtags ourselves based on presumption. See section 2 regarding Materials and Methods.
How do you define caring a prematures babies in the NICU by the twitter. Please explain.
The purpose of the paper was to identify how twitter and associated tweets shared could provide a discourse regarding the ways in which Covid-19 may impact responses to prematurity. The paper sought to understand, not define, caring by using twitter data to explore parents’ and healthcare professionals’ experiences of caring for premature babies in the NICU during the COVID pandemic (see page 3).
Please add section about premature, NICU health services in new normal (during Covid19). Add more than 20 papers.
Given that the COVID-19 pandemic is ongoing, this remains a very new area of study, and one in which studies are still ongoing or not yet published. Within the 26 sources cited in the Introduction, we included the most recent and relevant information around the experiences of caring for babies born prematurely during the pandemic from the perspectives of parents and healthcare professionals. We did not include papers specifically focused on medical interventions as this was beyond the scope of the study.
Reviewer 3 Report
This article describes a research project that is well designed, created, and implemented.
I would like to thank the authors for the opportunity to read their manuscript.
The study on this topic is very interesting. The structure is clear and logical and challenging.
All the required sections are presented with comprehensive details.
The Introduction, Methodology, Results, and Discussions sections are clearly presented and directly related to the objective of this research project.
Recommendations. I suggest:
- charts can be very beneficial to represent the results of the work.
- a grammar and spelling review.
- to authors add the following paper to the reference list: Fedushko S., Shakhovska N., Syerov Yu. Verifying the medical specialty from user profile of online community for health-related advices. CEUR Workshop Proceedings. Vol. 2255: Proceedings of the 1st International workshop on informatics & Data-driven medicine (IDDM 2018), Lviv, Ukraine, November 28–30, 2018. P. 301–310.
Results: Perhaps it is better to visualize in more charts. The results of this investigation are really significant.
And finally, the results, discussion, and conclusion were equally well-driven.
The research is timely and worthwhile. The authors provide fresh insight into the field.
Good luck with your research.
Author Response
This article describes a research project that is well designed, created, and implemented.
I would like to thank the authors for the opportunity to read their manuscript.
The study on this topic is very interesting. The structure is clear and logical and challenging.
All the required sections are presented with comprehensive details.
The Introduction, Methodology, Results, and Discussions sections are clearly presented and directly related to the objective of this research project.
Thank you so much for this positive feedback and for your engagement with the paper.
Recommendations. I suggest:
charts can be very beneficial to represent the results of the work.
Thank you for this excellent suggestion. We have added a table on pages 4-5 detailing the themes and exemplar quotes (tweets).
a grammar and spelling review.
We have read through the manuscript and made all required revisions.
to authors add the following paper to the reference list: Fedushko S., Shakhovska N., Syerov Yu. Verifying the medical specialty from user profile of online community for health-related advices. CEUR Workshop Proceedings. Vol. 2255: Proceedings of the 1st International workshop on informatics & Data-driven medicine (IDDM 2018), Lviv, Ukraine, November 28–30, 2018. P. 301–310.
Thank you for suggesting this interesting paper. However, as the aim of the study was not seeking to address verification of health advice, after careful consideration we felt this paper was not sufficiently relevant to the findings of this paper.
Results: Perhaps it is better to visualize in more charts. The results of this investigation are really significant.
See above. Thank you for this suggestion.
And finally, the results, discussion, and conclusion were equally well-driven.
The research is timely and worthwhile. The authors provide fresh insight into the field.
Good luck with your research.
Thank you so much for your support of the paper.
Round 2
Reviewer 2 Report
Dear Authors,
Thank you for clearly explanation. Good luck for your manuscript.
Author Response
Thank you so much for your kind words and thoughtful reviews.
Reviewer 3 Report
I've checked the revised version and I recommend its acceptance. Congrats.
Author Response
Thank you so much for your good wishes and thoughtful reviews.